# A Computational Toolbox to Investigate the Metabolic Potential and Resource Allocation in Fission Yeast

Pranas Grigaitis,[a] Douwe A. J. Grundel,[a]* Eunice van Pelt-KleinJan,[a,b] Mirushe Isaku,[a] Guixiang Xie,[a] Sebastian Mendoza Farias,[a] Bas Teusink,[a] Johan H. van Heerden[a]

aSystems Biology Lab, Amsterdam Institute of Molecular and Life Sciences (AIMMS), Vrije Universiteit Amsterdam, Amsterdam, The Netherlands
bTiFN, Wageningen, The Netherlands

**ABSTRACT** The fission yeast, *Schizosaccharomyces pombe*, is a popular eukaryal model organism for cell division and cell cycle studies. With this extensive knowledge of its cell and molecular biology, *S. pombe* also holds promise for use in metabolism research and industrial applications. However, unlike the baker's yeast, *Saccharomyces cerevisiae*, a major workhorse in these areas, cell physiology and metabolism of *S. pombe* remain less explored. One way to advance understanding of organism-specific metabolism is construction of computational models and their use for hypothesis testing. To this end, we leverage existing knowledge of *S. cerevisiae* to generate a manually curated high-quality reconstruction of *S. pombe*'s metabolic network, including a proteome-constrained version of the model. Using these models, we gain insights into the energy demands for growth, as well as ribosome kinetics in *S. pombe*. Furthermore, we predict proteome composition and identify growth-limiting constraints that determine optimal metabolic strategies under different glucose availability regimes and reproduce experimentally determined metabolic profiles. Notably, we find similarities in metabolic and proteome predictions of *S. pombe* with *S. cerevisiae*, which indicate that similar cellular resource constraints operate to dictate metabolic organization. With these cases, we show, on the one hand, how these models provide an efficient means to transfer metabolic knowledge from a well-studied to a lesser-studied organism, and on the other, how they can successfully be used to explore the metabolic behavior and the role of resource allocation in driving different strategies in fission yeast.

**IMPORTANCE** Our understanding of microbial metabolism relies mostly on the knowledge we have obtained from a limited number of model organisms, and the diversity of metabolism beyond the handful of model species thus remains largely unexplored in mechanistic terms. Computational modeling of metabolic networks offers an attractive platform to bridge the knowledge gap and gain new insights into physiology of lesser-studied organisms. Here we showcase an example of successful knowledge transfer from the budding yeast *Saccharomyces cerevisiae* to a popular model organism in molecular and cell biology, fission yeast *Schizosaccharomyces pombe*, using computational models.

**KEYWORDS** fission yeast, genome-scale model, resource allocation

The fission yeast *Schizosaccharomyces pombe* is a popular eukaryal model organism for cell division and cell cycle studies. With this extensive knowledge of its cell and molecular biology, *S. pombe* also holds promise for use in metabolism research and industrial applications. However, unlike the baker's yeast *Saccharomyces cerevisiae*, a major workhorse in these areas, cell physiology and metabolism of *S. pombe* remain much less explored. While these two yeasts share some similarities, distinct differences in, e.g., cell cycle regulation (1), mode of cell division (2), glucose transport (3) and utilizable carbon sources (4), make *S. pombe* a highly complementary model for studies into eukaryotic metabolism. A deeper understanding of *S. pombe* metabolism, therefore, offers opportunities to expand

Address correspondence to Pranas Grigaitis, p.grigaitis@vu.nl, or Johan H. van Heerden, j.van.heerden@vu.nl.

*Present address: Douwe A. J. Grundel, Molecular Systems Biology, Groningen Biomolecular Sciences and Biotechnology Institute, University of Groningen, Groningen, The Netherlands.

The authors declare no conflict of interest.

our knowledge of the larger eukaryal metabolic landscape. In this regard, computational approaches can provide a useful means to leverage the extensive metabolic knowledge from *S. cerevisiae* to explore *S. pombe* metabolism.

Computational approaches have become increasingly important to unravel and understand metabolism in diverse species, ranging from bacteria to humans. Arguably the most successful approaches in both applied and fundamental research are based on genome-scale metabolic models (GEMs) (5). A GEM is a computable knowledgebase that is essentially a compendium of all reactions of an organism: its metabolic potential, based on the genome sequence. GEMs have successfully been applied in diverse settings, including the metabolic engineering of microorganisms (6, 7), studies of human diseases or disease-causing pathogens (8, 9), drug development (10), and the investigation of interactions within microbial communities (11). Furthermore, by providing a general framework based on the genome sequence of an organism, GEMs allow for efficient transfer of metabolic knowledge between organisms.

GEMs of *S. pombe* have previously been constructed. However, several issues, including incompatibility with current Systems Biology Markup Language (SBML) standards (12, 13), a lack of gene-protein-reaction (GPR) associations, or automated reconstruction without additional curation (12, 14), significantly limited their utility. Furthermore, recent extensions of the GEM framework to include regulation and resource allocation dynamics now enable the exploration of complex metabolic behaviors such as the Crabtree-effect (analogous to the Warburg-effect seen in human cells) that cannot be explained with conventional GEMs.

Thus, in this study, we exploited the extensive metabolic knowledge and modeling toolset available for *S. cerevisiae* to generate an updated computational toolbox for *S. pombe*, consisting of a genome-scale metabolic model, *pomGEM*, and a resource allocation model, *pcPombe*. We manually curated and calibrated both models using published experimental data. We used the *pcPombe* model to identify proteome constraints that dictate the growth and metabolic strategy of *S. pombe* in glucose-limited chemostat cultures. We found that behavior appears to be governed by constraints similar to those operating in *S. cerevisiae*. These models provide essential tools to further expand knowledge of *S. pombe's* metabolism, specifically, and eukaryotic metabolism in general.

## RESULTS

**Reconstruction of the *S. pombe* metabolic network.** We first aimed to create a manually curated, high-quality reconstruction of the *S. pombe* metabolic network. Therefore, we coupled automated reconstruction tools (using *Saccharomyces cerevisiae* metabolic reconstruction *Yeast8.3.3* model [15] as a template) with thorough manual curation (see Materials and Methods) to construct the *pomGEM*, a manually curated GEM of *S. pombe* (Fig. 1A) that meets current standards for annotation and reusability. Manual curation of newly reconstructed GEMs is critical for accurate prediction of metabolic phenotypes. For example, during the curation we removed the reactions of the glyoxylate cycle, a pathway that is active in *S. cerevisiae* but absent in *S. pombe* (4), and the reason why *S. pombe* cannot utilize two-carbon compounds for growth. In addition, we replaced the biomass objective function (BOF) of the *Yeast8.3.3* model with the BOF used in the SpoMBEL1693 model (13), which is based on experimental measurements of *S. pombe* (Fig. 1B).

Next, we looked at the energetic parameters. First, we confirmed that the P/O ratio (ATP produced per oxygen atom reduced) in the model is 1.28, consistent with experimental measurements (16). In terms of ATP maintenance parameters, we kept the non-growth-associated ATP maintenance (NGAM) demand at 0.7 $mmol\ gDW^{-1}\ h^{-1}$ from the *Yeast8.3.3* model, in agreement with experimentally determined values for *S. pombe* (0.66–0.83 $mmol\ gDW^{-1}\ h^{-1}$) (16). Furthermore, we estimated the growth-associated ATP maintenance (GAM) value (Fig. 1C). We used published experimental measurements of growth yield on glucose ($Y_{X/S}$) in fully respiratory glucose-limited cultures of *S. pombe* and varied the GAM value to achieve the target yield $Y_{X/S} = 0.432\ g\ biomass\ (g\ glucose)^{-1}$. The target $Y_{x/s}$ corresponded to $GAM = 58.3\ mmol\ gDW^{-1}$, comparable with 55.3 $mmol\ gDW^{-1}$ in the *Yeast8.3.3*

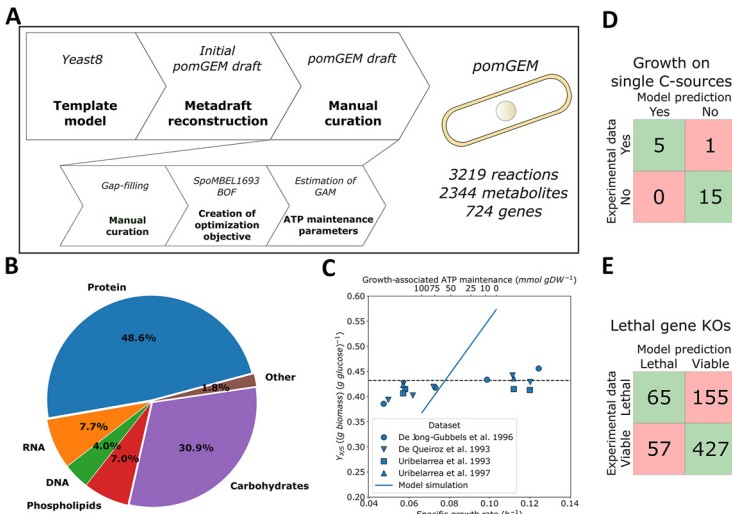

**FIG 1** Reconstruction of the *pomGEM*, the genome-scale metabolic model of *S. pombe*. (A) The workflow of the reconstruction. (B) The composition of *S. pombe* biomass, defined in the *pomGEM*. (C) Estimation of the GAM value. Glucose uptake flux was fixed to 1.0 *mmol* $gDW^{-1}$ $h^{-1}$ and the maximal specific growth rate $\mu$ (solid blue line) was predicted with varying GAM value. Growth yield on glucose $Y_{X/S}$ was computed based on the predicted specific growth rate. The target yield on glucose [$Y_{X/S} = 0.432$ *g biomass* $(g\ glucose)^{-1}$] (dashed horizontal line) was computed as an average of experimentally determined $Y_{X/S}$ from glucose-limited cultures with $D > 0.1\ h^{-1}$ (4, 16–18). (D and E) Benchmarking of the *pomGEM* model. (D) Prediction of growth on single carbon sources (experimental data from Choi et al. [19] and our measurements; see Table S1 for details); (E) Prediction of the lethality of single gene KOs (experimental data from Kim et al. [20]). BOF, biomass objective function; GAM, growth-associated maintenance; KO, knockout.

model. The *pomGEM* model showed very good agreement for the predicted flux values in central carbon metabolism with measured fluxes in glucose-limited chemostat cultures at $D = 0.1\ h^{-1}$ (21) (Fig. S1).

We benchmarked the *pomGEM* model by predicting growth on a panel of 21 single carbon sources (Fig. 1D, Table S1) and lethality of single-gene knockouts (KOs; Fig. 1E, Table S2). Predictions of growth on single carbon sources were correct for all carbon sources except one, ribose: Choi et al. (19) reported growth on ribose, but *pomGEM* predicted no growth (false negative). It should be noted that the growth medium used for testing in the study by Choi et al. (19) is not clearly defined, as such it cannot be unambiguously concluded that this strain can grow on d-ribose as the sole carbon source. Of the predicted phenotypes, 69.9% of single-gene KOs were true predictions (match between model and experimental data) for the entire data set, while false positives (viable only *in silico*) and false negatives (viable only *in vivo*) were 22.0% and 8.1% of the data set, respectively. We, however, were not able to test the single-gene KOs on previously published reconstructions due to inherent technical issues with these models.

We also performed a check on the reaction essentiality to compare the prediction accuracy with the *SpoMBEL1693* model, where essentiality was assessed in terms of reactions rather than genes. We determined the essentiality (see Materials and Methods) of 2,017 model reactions with gene-protein-reaction (GPRs) associations and mapped the GPRs with the individual genes in the data set of gene KOs (Table S3). *pomGEM* showed a true prediction rate of 74.7%, a good improvement (13.5%) on the true prediction rate achieved by *SpoMBEL1693* reconstruction (61.2%) (13).

**Development of the proteome-constrained model of *S. pombe*.** FBA-based models are powerful tools to investigate the potential of metabolic networks, but the ground assumptions of the method limit the prediction of metabolic phenotypes. As a rule, FBA predictions will identify the metabolic strategy that leads to the highest biomass yield on the limiting nutrient when optimization objective is maximization of growth rate. For instance, under glucose-limited conditions, a GEM of *S. cerevisiae* will always predict a high-yield ATP production strategy, complete respiration of glucose to $CO_2$ and water. In reality, cells will

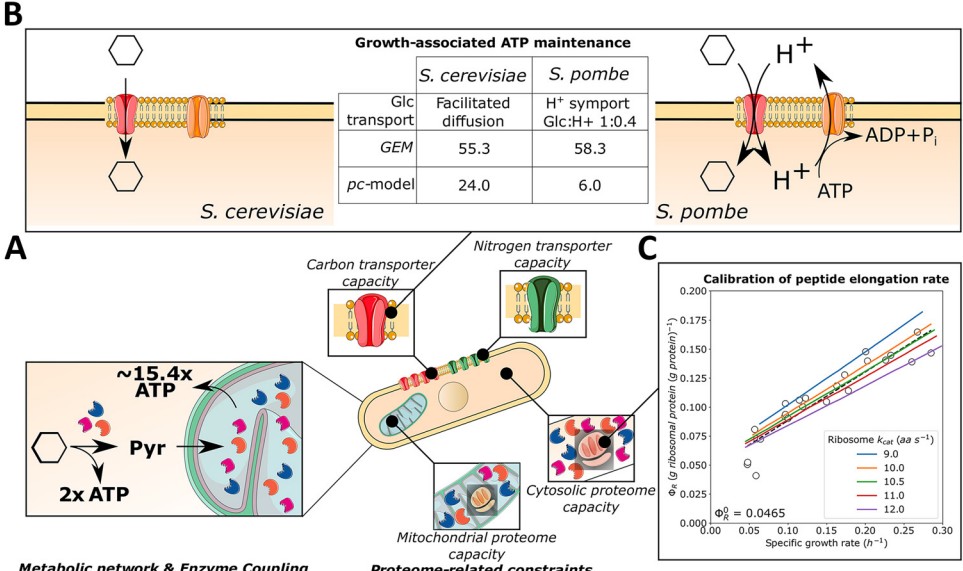

**FIG 2** Calibration of the proteome-constrained model of *S. pombe*, *pcPombe*. (A) The representation of different layers of the *pcPombe* model. The metabolic model (*pomGEM*) is complemented with a fine-grained description of protein turnover (reactions of protein translation, folding, degradation, and dilution by growth) and a set of compartment-specific proteome constraints (corresponding to proteome capacity of plasma membrane, mitochondria, and cytosol). (B) Representation of the glucose transport in *S. cerevisiae* and *S. pombe*, and the estimates of ATP maintenance costs for both organisms. (C) Calibration of the peptide elongation rate. The "inactive" fraction of ribosomes ($\Phi_R^0$) was estimated from the experimental data (black dashed line, linear fit of the experimental points), and growth on varying levels of glucose was simulated with different ribosome $k_{cat}$ values. Glc, glucose.

switch to fermentation, a lower ATP-yield strategy, when the glycolytic flux increases beyond a certain threshold. Thus, metabolic phenotypes that do not correspond to the highest-yield strategy cannot be predicted with FBA unless additional constraints are added that reflect physiological constraints (22).

An important constraint relates to the allocation of limited cellular resources. If metabolic reaction-associated protein costs are accounted for, different condition-dependent modes of growth, e.g., the switch between respiration and fermentation (23), can be reproduced. GEMs therefore can be improved by introducing the concept of resource allocation: optimal partitioning of the limited resources among the metabolic processes, based on the costs of energy and biosynthetic resources (e.g., amino acids) needed for implementing each metabolic pathway. Over the last 15 years, different extensions of GEMs were proposed in order to predict optimal resource allocation in different microorganisms (24). Recently, we introduced a proteome-constrained (*pc-*) model of *S. cerevisiae* (*pcYeast*) (25) that can accurately predict low and high biomass yield strategies under different growth conditions. In a similar spirit, we constructed *pcPombe*, a proteome-constrained model of *S. pombe*, based on the *pomGEM* model (Fig. 2A).

The *pcPombe* model (model explained in detail in the Text S1) captures the interplay of metabolism and cellular resource allocation by (i) coupling metabolic processes with respective protein demand, and (ii) coupling protein abundance with compartment-specific proteome capacity constraints. Thus, we first extended the metabolic model by introducing fine-grained descriptions of protein turnover (reactions protein synthesis, folding, degradation, and dilution by growth). Then, we compiled data from literature and/or specialized biological databases (see Materials and Methods and Text S1) to parametrize the *pcPombe* model (e.g., $k_{cat}$ values, Fig. S2) and establish compartment-specific proteome constraints with *pcYeast* as template (25). We then further calibrated the *pcPombe* model with available experimental data, as explained below.

**Calibrating ATP maintenance and protein translation costs in *pcPombe*.** A substantial amount (~40% in *S. cerevisiae* [26]) of ATP maintenance costs can be explained by protein turnover processes. As these processes are now modeled explicitly in the *pcPombe*

model, we used the measurements of biomass yield on glucose (Fig. 1C), to determine the GAM value for the *pcPombe* model (Fig. 2B). We first explicitly split the ATP maintenance into two components, cytosolic and mitochondrial ATP maintenance (GAM and mitoGAM, respectively). We based this decision on the fact that mitochondria are special organelles; they have a circular genome that stores a small number of protein-coding genes, and the translate them using a distinct mitochondrial pool of ribosomes. In the model, the exact number of mitochondria per cell is not specified, therefore a practical way to express the maintenance costs is mmol ATP per gram of mitochondrial protein.

Although protein turnover cost is a major determinant of GAM, other processes, which are often not explicitly modeled, can significantly influence this value. For example, in *S. cerevisiae*, glucose enters the cell via facilitated diffusion, while di- or oligosaccharides (maltose, maltotriose, raffinose, etc.) are imported into the cell through sugar: $H^+$ symport, leading to additional energetic costs of using these sugars for growth (27). However, in *S. pombe*, glucose transporters are also sugar: $H^+$ symporters, with a stoichiometry of 1:0.4 for glucose and protons, respectively (3). We have thus modified the stoichiometry of glucose import reactions in both *pomGEM* and *pcPombe* to reflect this.

The actual energetic costs here come from the fact that the protons, imported with the sugar, must be pumped out of the cell by the plasma membrane $H^+$-ATPases to maintain the proton balance in the cell. If this energetic cost of glucose transport is not accounted for, the growth rate will be significantly overestimated, especially during respiratory growth when the mitochondrion is used, and this is a consequence of two factors. First, by neglecting consumption of ATP by the $H^+$-ATPase, more ATP will be available for growth; in the model, correctly predicting the growth yield will then require a much higher GAM value. Second, increased cytosolic proton availability in the model will drive increased mitochondrial ATP synthase activity, leading to a higher ATP yield, and hence a higher estimated GAM value. Therefore, we added an additional constraint to the *pcPombe* model that couples glucose import to $H^+$-export through plasma membrane $H^+$-ATPases (see discussion of this modeling step in Text S1 1.4), thereby preventing incorrect use of these protons. With this additional constraint, we then estimated the ATP maintenance value.

While the GAM values for the metabolic models of *S. cerevisiae* and *S. pombe* were very similar, modification of the glucose transport mechanism resulted in a significant difference in the GAM values of the respective proteome-constrained models. In the end, we determined values of 6 $mmol\ gDW^{-1}$ and 6 $mmol\ (g\ mitochondrial\ protein)^{-1}$ for GAM and mitoGAM, respectively (Fig. 2B). The estimated GAM value for *pcPombe* is thus considerably smaller than the one for *pcYeast* (24 $mmol\ gDW^{-1}$) once the additional energetic costs of glucose transport is accounted for (Fig. 2B). For mitoGAM, the same value (6 $mmol\ (g\ mitochondrial\ protein)^{-1}$) was used in both *pcYeast* and *pcPombe*.

Next, we assessed the peptide elongation rate of the cytosolic ribosomes and the fraction of the proteome occupied by "inactive" ribosomes ($\Phi_R^0$) following Metzl-Raz et al. (28) (Fig. 2C); we have shown that these two parameters are key for the *pcYeast* model predictions (25). We used quantitative proteomics data from turbidostat experiments in Edinburgh minimal media (EMM2) (2% glucose), supplemented with different single nitrogen sources (29). First, we computed the fraction of "inactive" ribosomes $\Phi_R^0 \approx 0.05\ g\ (g\ protein)^{-1}$ (95% confidence interval, excluding the cultures grown with tryptophan as a nitrogen source: 0.041–0.052) from the linear regression of the experimental data points (Fig. 2C, black dashed line). Notably, the fraction of the "inactive" ribosomes is around 40% lower in *S. pombe* than in *S. cerevisiae* ($\Phi_R^0 \approx 0.08$) (28). Next, we estimated the peptide elongation rate in *S. pombe*, a parameter never reported in the literature (to the best of our knowledge). Thus, we ran a set of model simulations, where we varied the peptide elongation rate $k_{cat,ribo}$ around the initial value of $k_{cat,ribo} = 10.5\ aa\ s^{-1}$ from *S. cerevisiae* (28) (Fig. 2C). We concluded that the value of 10.5 $aa\ s^{-1}$ showed the best agreement with the experimental data. This suggests that although *S. cerevisiae* and *S. pombe* diverted in their evolutionary tracks relatively long time ago, their ribosomes seem to have remained highly functionally conserved.

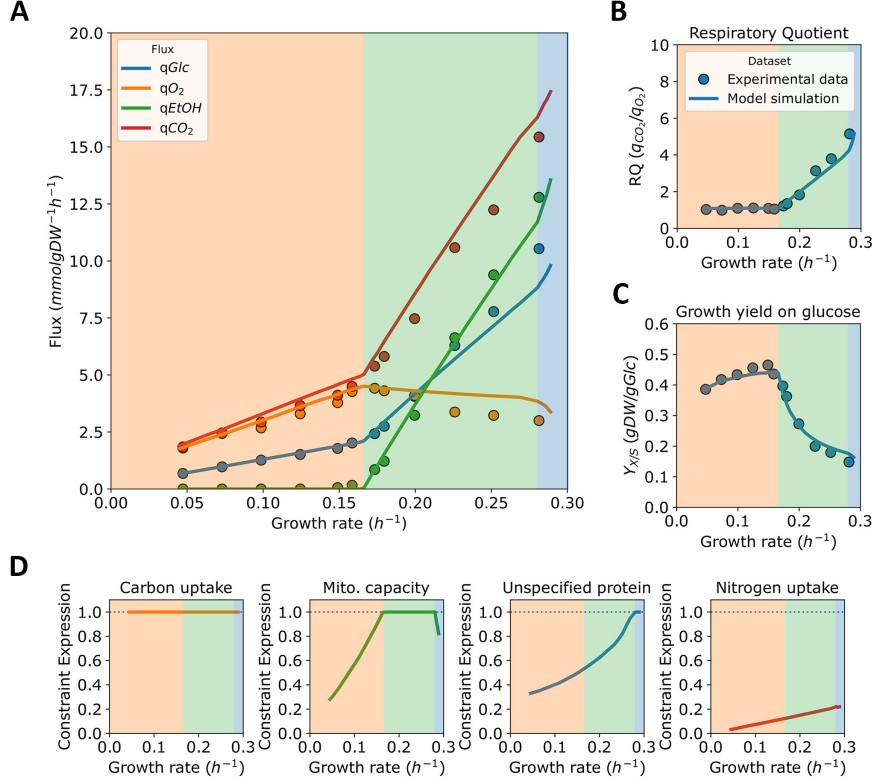

**FIG 3** Fluxes, physiological parameters, and active proteome constraints in glucose-limited growth of *S. pombe*. (A) Main predicted fluxes from glucose-limited chemostats. (B and C) Physiological parameters of the growth in glucose-limited chemostats. (B) Respiratory quotient, the ratio between the specific fluxes of carbon dioxide and oxygen; (C) Growth yield on glucose, the ratio between growth rate and glucose uptake. Experimental data (points) in panels A–C from de Jong-Gubbels et al. (4). (D) Active proteome constraints, predicted by the *pcPombe* model. Shading of different growth regimes in panels A–D corresponds to active proteome constraints, plotted in panel D.

**Identifying growth-limiting proteome constraints in glucose-limited chemostats.** The key feature of the *pcPombe* model is the ability to predict multiple facets of microbial physiology: flux distributions, proteome composition, and, most importantly, compartment-specific proteome constraints that actively limit the maximal growth rate. Therefore, as a use case example, we used the *pcPombe* model to identify the active constraints that drive the physiology of *S. pombe* growing in glucose-limited chemostats at an increasing dilution rate (Fig. 3).

We mimicked different extracellular glucose concentrations in the model by varying the saturation factor of the glucose transporters (Text S1) and used binary search (25) to find the maximal specific growth rate and corresponding flux distribution for every value of the saturation factor (Fig. 3A). The predicted fluxes, based on external metabolites, were also used to compute the physiological parameters (yield on glucose and the respiratory quotient) of cell cultures (Fig. 3B and C).

Based on the active compartment-specific proteome constraints (Fig. 3D), we partition the simulation (along the predicted specific growth rate) into three parts (shading in all the panels of Fig. 3). First, at very slow growth, the only active (i.e., the constraint expression equals 1 in Fig. 3D) proteome constraint is carbon uptake (carbon transporter capacity). Carbon transporter capacity remains the only active proteome constraint before the onset of ethanol formation (critical growth rate $\mu_{crit} = 0.16\ h^{-1}$), during which a second active proteome constraint is encountered—the mitochondrial proteome capacity (see below).

As growth rate continues to increase, the active constraints change (blue shaded region in Fig. 3), and so does the predicted metabolic behavior. At very fast growth rates, instead of mitochondrial proteome capacity, the unspecified protein (UP) fraction, starts to limit growth. The UP is a collective term that aggregates all proteins that do not contribute

directly to biomass synthesis (i.e., metabolically inactive proteins) into a single artificial protein of average composition and length. The minimal UP mass fraction is, therefore, a proxy for total cytosolic proteome capacity, which becomes an active constraint when the UP fraction in the proteome reaches the minimal value that we estimated based on proteomics data (29); at this minimal value, the cytosol is maximally filled with metabolically active proteins. As a result, any increase in growth must be accompanied by trading of mitochondrial proteins for cytosolic ones (Fig. 3D, "Mito. capacity" panel). Both the minimal UP fraction and the maximal mitochondrial proteome capacity (Text S1) are estimated parameters, due to lack of supporting experimental data. We, however, believe that the sequence of active proteome constraints (thus, also the fitted parameter values) is supported by literature data, coming from both *S. cerevisiae* and *S. pombe*.

First, we addressed the mitochondrial capacity being the constraint behind the onset of ethanol formation. We tested our claims by increasing the minimal UP fraction to the level that sets the UP minimum to be hit at $\mu^*_{UP\ hit} = 0.16\ h^{-1}$ $(= \mu_{crit})$, and the glucose transporters were fully saturated and mitochondrial capacity constraint was relaxed. The flux predictions we acquired were considerably different from the experimental data of de Jong-Gubbels et al. (4); a rapid increase in ethanol production was observed as the UP minimum was hit, and the maximal growth rate was $\mu_{max} = 0.18\ h^{-1}$. We concluded that the flux profile at the maximal growth rate $\mu_{max} = 0.18\ h^{-1}$ (which resembled experimental measurements at $\mu = 0.29\ h^{-1}$), was highly unlikely to be correct, and therefore we discarded such scenario. Next, we considered the active constraint (UP minimum) for growth in in glucose excess. Malina and colleagues (30) determined that both *S. cerevisiae* and *S. pombe* allocate a very similar fraction (and in both cases small, <5%) of the proteome to TCA cycle and oxidative phosphorylation proteins. This suggests that the same constraints limit growth in glucose excess, and we have previously shown that this constraint is the cytosolic proteome capacity (25). Therefore, the active constraints at slower growth (onset of ethanol formation) must be of a different nature, and knowledge of *S. cerevisiae* again pointed to mitochondrial proteome capacity as the constraint limiting growth at that phase. We constructed the *pcPombe* model with these observations with *S. cerevisiae* in mind, and since we achieved a good flux prediction, we argue that it is *the* active constraint under this growth regime.

When the predicted growth rate approaches the maximal predicted growth rate, growth is no longer limited by carbon transporter capacity, and thus, only one constraint (minimal UP mass fraction) remains active. In this state, excretion of additional overflow products (e.g., pyruvate) is predicted, consistent with the behavior of *S. cerevisiae* at glucose excess conditions. It should be noted that the predicted maximal growth rate in the EMM2 medium $(\mu_{max} = 0.29\ h^{-1})$ is dependent on the minimal UP fraction in the proteome, a parameter we fit. However, we argue that our estimate is reasonable, since *pcPombe* correctly predicts the maximal growth rate on the rich yeast extract with supplements (YES) medium with the same parameter values $(\mu_{max} = 0.34\ h^{-1})$ (31). To summarize, here, we used the *pcPombe* model together with the existing knowledge on *S. cerevisiae* to verify the identity of proteome constraints, which actively limit growth in a condition-dependent manner.

**Maximal growth rate of *S. pombe* is defined by limited proteome access.** We observed that the maximal experimentally determined growth rate of *S. pombe* in a minimal medium $(\mu_{max} = 0.30\ h^{-1})$ is substantially lower than the maximal growth rate of the *S. cerevisiae* CEN.PK strain (Verduyn medium [32] with glucose as carbon source; $\mu_{max} = 0.40\ h^{-1}$ [25]). We speculate that the lower maximal growth rate is an outcome of lower protein density in *S. pombe* biomass, and *S. cerevisiae* has a "higher budget" to accommodate proteins, needed for faster growth. *S. pombe* exhibits a constant protein density of $0.43\ g\ (gDW)^{-1}$ (4), while in *S. cerevisiae*, the respective value is growth rate-dependent and is reported to be $0.505$ $g\ (gDW)^{-1}$ at $\mu = 0.375\ h^{-1}$ (33). Although different in absolute amounts, similar proteome partitioning at the maximal growth rate suggests that the maximal growth is limited by similar constraints.

The design of the *pc*-models allows for the inspection of proteome allocation in a fine-grained manner; for every enzyme that supports growth by a catalyzing a metabolic flux, a corresponding *minimal* protein demand can be computed for the (hypothetical) case that

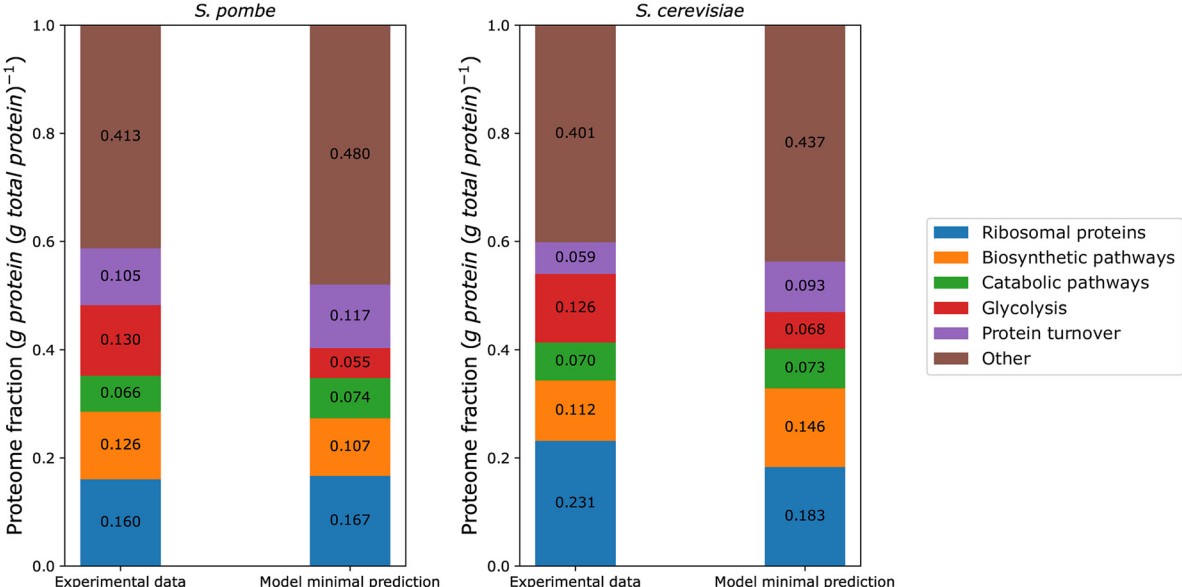

**FIG 4** Proteome composition of *S. pombe* and *S. cerevisiae* at maximal growth rate. Experimentally measured proteome composition (left bars) and predicted minimal protein level (right bars) represented as proteome mass fractions, in $g\ (g\ protein)^{-1}$. Experimental data for both *S. pombe* and *S. cerevisiae* were taken from Malina et al. (30), and model predictions for *S. cerevisiae* were taken from Elsemman et al. (25). Experimentally determined proteome composition in the figure corresponds to the average of measurements reported in Malina et al. (30).

all proteins work at their maximal rate: $v = [e_i] \times k_{cat,i}$. At slow growth with low metabolic fluxes, the minimal protein demand will be low. Typically, under these conditions, cells express metabolic proteins at higher levels compared to the minimal predicted protein demand (25, 34). Yet, the difference decreases with increasing growth rate for *S. cerevisiae* (25), with a major exception of ribosomal proteins (because ribosomal parameters are fitted explicitly; Fig. 2C). To illustrate the predicted proteome partitioning, we looked into the predictions of *pcPombe* at the maximal predicted growth rate and compared the minimal predicted protein demand with experimental data (30) (Fig. 4).

We used a manually curated proteome annotation set (Table S3) to map proteins to different functional groups or pathways. To avoid comparing >30 pathways with small proteome fractions, we grouped pathways into a handful of coarse-grained clusters (Fig. 4), with the exception of glycolysis, which is directly compared as a single pathway instead of being lumped with the rest of the catabolic (e.g., pentose phosphate pathway, TCA cycle, and oxidative phosphorylation) proteins. For additional insights, we also considered the proteome composition of *S. cerevisiae* and compared it to that of *S. pombe*. Both models predicted and experimentally determined proteome fractions; most of these coarse-grained clusters occupy comparable-sized proteome fractions in both organisms. Also, the deviations between predicted minimal protein demand and experimental protein fraction have similar patterns in both organisms. When looking at predictions, a significant deviation from experimental data is seen in the proteome fraction involved in the metabolism of carbohydrates. The experimentally determined fraction of glycolytic enzymes is 2-fold higher than the predicted minimal demand.

This result is not completely surprising, since we observed a similar result (ca. 2-fold) in previously published proteome data of *S. cerevisiae* cultures at the maximal growth rate in minimal medium (batch cultures with excess glucose) (25). It appears, therefore, that both these yeasts have an overcapacity of glycolytic enzymes that is not needed to support the maximal growth rate; why this is the case, is currently not understood. Overall, we observed that the proteome partitioning at maximal growth is similar between *S. pombe* and *S. cerevisiae*. This supports the inference that the maximal growth under nutrient excess is limited by a similar constraint in both organisms. Following the predictions of proteome-constrained models, we suggest that this constraint is total proteome capacity.

## DISCUSSION

In this study, we used metabolic modeling and data from the well-studied budding yeast, *S. cerevisiae*, to gain insights into the metabolism and physiology of the distantly related fission yeast, *S. pombe*. As a result, we presented a computational toolbox to investigate fission yeast metabolism at genome scale. Two types of models, in our view, are required to cover this need: a genome-scale metabolic model (metabolic potential) and a proteome-constrained (pc-) model (resource allocation).

Here, we first developed a manually curated and calibrated GEM, *pomGEM*, based on a metabolic model of budding yeast *S. cerevisiae* (15) (Fig. 1). As an outcome of the model calibration, in this manuscript we provide for the first time a comprehensive and data-supported estimate of growth-associated maintenance (GAM) costs of *S. pombe* (Fig. 1C). An earlier proposed GAM value of 17.37 $mmol\ gDW^{-1}$ (13) corresponds to an unrealistically high yield of biomass on glucose in aerobic settings, while our proposed value (58.3 $mmol\ gDW^{-1}$) corresponds well with existing experimental data. Moreover, the GAM value we estimated is very close to that reported for *S. cerevisiae* (55.3 $mmol\ gDW^{-1}$) (35), further supporting our estimate over previous estimates (13).

We benchmarked the *pomGEM* model by first predicting growth on single carbon sources (with only one false-negative, Fig. 1D), lethal single-gene KOs (Fig. 1E), and single-reaction KOs (Table S3). For the latter, the fraction of true predictions was approximately 75%, a good improvement on the previously reported model (61.2%) (13). In our study, we applied a rather stringent threshold for the viability of single-reaction KOs, considering the reaction essential if the predicted growth rate was below 90% of the wild-type value. We, thus, tested a different threshold (essential when the growth rate is zero) and arrived at effectively the same true prediction rate (74.7% vs 74.8% at zero growth threshold). This suggests that the overall performance of the *pomGEM* model in this regard is robust.

However, in the study by Sohn et al. (13), these authors of the *SpoMBEL1693* model reported an increase in the true prediction rate of up to 82.7% after significant manual curation. Here, the authors "reconciled" the false predictions, which arise from, e.g., duplicate reactions present in other compartments, or dead-end pathways, to achieve the higher true prediction rate. However, such an *ad hoc* approach requires supporting experimental data to resolve every false prediction reliably. Nonetheless, following the evolution of true prediction rates of the *S. cerevisiae* models (90.3% in *Yeast8* vs 83.6% in *Yeast4*) (36), in terms of genes, or the latest GEM of *E. coli* (>90%) (37), it is anticipated that with more experimental data, future iterations of *pomGEM* will similarly lead to further improvements in the true prediction rate.

On the basis of *pomGEM* and using *pcYeast* as template (25), we reconstructed (Fig. 2A) and calibrated (Fig. 2B and C) a proteome-constrained metabolic model of *S. pombe*, *pcPombe*. We first identified a major ATP maintenance component: plasma membrane $H^+$-ATPase activity, required to export protons that are imported through glucose/$H^+$ symport (Fig. 2B). We also estimated the peptide elongation rate of cytosolic ribosomes and found it to be similar to the rate reported for *S. cerevisiae* (Fig. 2C).

We used the *pcPombe* model to simulate the physiology of *S. pombe* in glucose-limited chemostats at different dilution rates (Fig. 3) and identified proteome constraints that actively limit growth. Despite a large evolutionary distance, constraints similar to those recently described for *S. cerevisiae* (25) were shown to dictate growth behaviors, with a mitochondrial proteome capacity limitation ultimately driving a switch from respiration to fermentation. Finally, we looked at the predicted minimal proteome demand at the maximal growth rate of *S. pombe* in minimal medium and compared it to experimental measurements (Fig. 4). For many coarse-grained proteome clusters, minimal predicted demands were comparable, and the prediction outcome was similar to that of *S. cerevisiae* at maximal growth rate in minimal medium. Such agreement suggests that the growth in nutrient excess is limited by similar constraints in both organisms, in this case, total proteome capacity constraint. A notable exception in predicted minimal demand versus experimental data was seen for glycolysis, where an experimentally determined proteome fraction was 2-fold higher than the minimal predicted demand. This result suggests a large over-capacity

of glycolytic enzymes, also found for *S. cerevisiae* (25). However, the reason for this over-capacity remains to be resolved.

Quantitative differences in proteome composition, especially at individual protein level, between the model and experimental measurements (likewise large or small), can be influenced by several factors. First, we consider the minimal protein demand in the model. This assumption ignores any preparatory protein expression, and the predicted protein abundance is highly dependent on the $k_{cat}$ values. The effects of other kinetic factors are also not accounted for, e.g., suboptimal saturation of enzymes and feedback effects (positive and negative alike) in the biochemical pathways. Therefore, protein "underutilization" (or "reserve capacity") is a frequently observed prediction of resource allocation models (25, 34). Second, GEMs consider only proteins with direct metabolic function (plus those directly related to protein turnover, in the *pcPombe* model). Thus, some proteins will be unaccounted for when mapping them to annotated pathways. Improved GPR annotations in future versions of *pomGEM* would reduce such "lost" mappings.

Throughout the manuscript, we considered very few applications of the computational toolbox, and only a handful of data sources. This is because the predictive power of the current *pomGEM* and *pcPombe* models is severely hampered by a lack of consistent, high-quality experimental data sets needed to calibrate and validate the models. The hope is that our current effort to provide a computational tool to study *S. pombe's* metabolism will stimulate an iterative cycle of hypothesis generation, experimental testing, and model refinement. For *S. cerevisiae*, its genome-scale model is already in its 8th iteration, with efforts beginning almost two decades ago (35). Throughout the years, essential modeling parameters, such as the GAM value (35), growth rate-dependent biomass composition (33), ribosome peptide elongation rate (28), and a large panel of kinetic parameters (15, 38), were determined. Thus, by aggregating a vast amount of existing literature data, and acquiring new experimental data sets (physiological data and proteomics), a proteome-constrained model of *S. cerevisiae* (*pcYeast*) was created and could be successfully tested in a number of scenarios, as seen in studies by Elsemman et al. (25) and Grigaitis et al. (unpublished).

Existing experimental data sets of *S. pombe*, unfortunately, are not as comprehensive. Although many of the data sets are of high-quality, they consider only one aspect of cell growth, for instance, exometabolite fluxes (4), or proteome composition (29). For modeling purposes, systemic experiments, which cover several layers of information at once (e.g., sampling from the same cultures to quantify bulk biomass composition, exometabolite fluxes, and proteome composition), as well as testing current predictions on active proteome constraints by, e.g., titrating expression of nonfunctional proteins targeted to specific cell compartments (e.g., cytoplasm, cell membrane, etc.), as has been done for *E. coli* (39), or by testing optimal protein allocation with evolution experiments (as performed in *Lactococcus lactis* [40]) will be extremely useful. Performing such experiments and subsequent model refinements will have great influence on the predictive power of the *pomGEM* and *pcPombe* models and will pave the way toward deeper understanding of metabolism and resource allocation of fission yeast *Schizosaccharomyces pombe*.

Lastly, recent studies suggested *S. pombe* could find novel applications in biotechnology, including winemaking (41) and flavor formation during food fermentations (42), but also as a possible cell factory (43). *S. pombe's* ability to grow in environments with low water activity, high alcohol content, very low pH, and a wide range of temperatures (44) make it an attractive, and perhaps underutilized, biotechnological tool. However, identifying metabolic engineering targets and predicting outcomes is a major challenge without a robust computational framework. The *pomGEM* model we present here, therefore, is a powerful tool that can be used to efficiently explore, *in silico*, *S. pombe's* metabolic potential, to identify metabolic engineering targets, and to design and optimize medium for different applications. These analyses can be complemented by studies with *pcPombe*, directed at the metabolic and physiological determinants of growth behavior under different growth conditions.

## MATERIALS AND METHODS

**Determination of growth on different carbon sources.** *Schizosaccharomyces pombe* strain CBS1042 (Westerdijk Fungal Biodiversity Institute, The Netherlands) was used to determine growth capacity on

different individual carbon sources. Glycerol stocks were prepared from cells grown to saturation in yeast extract peptone dextrose (YPD) medium and stored at −80°C. All cultures were performed at 30°C using EMM2 (45) as a base medium. All carbon source concentrations are expressed as carbon mol (C-mM) and were added to a final concentration of 600 C-mM (e.g., 100 mM glucose, 50 mM sucrose, 200 mM pyruvate, etc.). Growth experiments were carried out using a SpectraMax Plus 384 microplate reader (Molecular Devices, Silicon Valley, California). A standardized procedure was used for revival and inoculation of cultures. Briefly, glycerol stocks were revived by 100× diluted inoculation into EMM2 with 600 C-mM glucose. After approximately 7 h, overnight cultures were again diluted and inoculated into EMM2 + glucose to a final $OD_{600}$ of 0.02. The next day, fresh media containing the carbon sources to be tested (Table S1) were inoculated to a final $OD_{600}$ of 0.01. After 6 h, cultures were again diluted (final $OD_{600}$ of 0.01) using the same medium and transferred to 96-well microtiter plates. Per carbon source, 10 technical replicates were included (300 $\mu$L per well), along with 5 negative controls (growth medium with carbon source, no cells). Temperature was set to 30°C and double orbital shaking at 600 rpm was used. OD values were recorded in 5-min intervals at 600 nm for approximately 80 h.

**Reconstruction of the metabolic network of *Schizosaccharomyces pombe*.** The metabolic network of *S. pombe* was reconstructed with CBMPy MetaDraft (46), using the reference proteome sequence from PomBase (47) and *Yeast8.3.3* (15) as the template model. Model simulations, as well as manual refinement and gap-filling were performed in CBMPy 0.8.2 (48) under the Python 3.9 environment with IBM ILOG CPLEX 20.10 as the linear program (LP) solver.

**Mapping essential reactions to gene lethality.** Essential reactions in the model were determined by computing the predicted growth rate with a single reaction being blocked (lower and upper flux bounds set to 0.0) for all reactions in the model. If blocked flux through a reaction resulted in a predicted growth rate 90% or lower of the maximal (wild-type) growth rate, we considered such reaction essential; otherwise, the mutant is considered viable. Only reactions with existing gene-protein-reaction (GPR) associations were considered and compared with experimental data. For GPRs containing an "OR" clause, the experimentally determined essentiality must match for all listed genes (or combinations of) to be assigned either "viable" or "essential." For GPRs containing an "AND" clause, reaction was assigned "essential" if at least one of the genes was experimentally determined to be essential; "viable" was assigned the same way as for "OR" clauses. Conflicting results or missing essentiality experiments were labeled "ambiguous" and not considered further.

**Reconstruction and simulations of the proteome-constrained model.** The detailed description of reconstruction of the proteome-constrained model of *S. pombe* is provided in the Text S1. We used the reference proteome of *S. pombe* from UniProt (49). The kinetic data (enzyme turnover values) were collected from the BRENDA database (50). For every enzymatic complex with an Enzyme Commission (EC) number, we queried the BRENDA database for a value from the wild-type enzymes. When available, values from *S. cerevisiae* or *S. pombe* were preferentially selected. Otherwise, the highest value for a wild-type enzyme in mesophilic conditions (and close-to-growth conditions of *S. cerevisiae* or *S. pombe*) was taken. When no $k_{cat}$ value was available, we assumed $k_{cat} = 50\ s^{-1}$ as a default value (close to the median $k_{cat}$, Fig. S2). If the experimentally determined $k_{cat}$ value was lower than $2/3\ s^{-1}$, we set this value.

5'-UTR sequences and proteome annotations (composition of macromolecular complexes, Gene Ontology terms, etc.) were collected from PomBase (47). The *pcPombe* model was simulated using CBMPy 0.8.2 (48) under the Python 3.9 environment with IBM ILOG CPLEX 20.10 and SoPlex 4.0 (51) as the low- and high-precision LP solver, respectively.

**Data availability.** Experimental data on growth of *S. pombe* on different carbon sources is provided in Table S1. The *pomGEM* model, *pcPombe* model, and the materials used to generate the *pcPombe* model, together with information required to generate the figures of this manuscript, are available on Zenodo https://doi.org/10.5281/zenodo.6513462.

## SUPPLEMENTAL MATERIAL

Supplemental material is available online only.

**TEXT S1**, PDF file, 0.1 MB.
**FIG S1**, EPS file, 0.2 MB.
**FIG S2**, EPS file, 0.05 MB.
**TABLE S1**, XLS file, 0.03 MB.
**TABLE S2**, XLS file, 0.1 MB.
**TABLE S3**, XLS file, 0.4 MB.

## ACKNOWLEDGMENTS

We thank Istvan T. Kleijn (Imperial College London, the United Kingdom) for sharing unpublished transcriptomics and proteomics data and discussions, Brett G. Olivier for his help on the modeling software, and Julius Battjes for discussions. We thank SURFsara for the HPC resources through access to the Lisa Compute Cluster.

P.G. and B.T. acknowledge support by Marie Skłodowska-Curie Actions ITN "SynCrop" (grant agreement no. 764591). E.V.P.-K. and B.T. acknowledge funding from the Netherlands Organization for Scientific Research (grant no. ALWTF.2015.4).

The project is partly organized by and executed under the auspices of TiFN, a public-private partnership on precompetitive research in food and nutrition. Among other sources declared, funding for this research (to E.V.P.-K. and B.T.) was obtained from Friesland Campina, CSK Food Enrichment and The Netherlands Organization for Scientific Research.

The authors have declared that no competing interests exist in the writing of this publication.

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
