## [Reviewer comments · mSystems]

A computational toolbox to investigate the metabolic potential and resource allocation in fission yeast

Pranas Grigaitis, Douwe Grundel, Eunice van Pelt-KleinJan, Mirushe Isaku, Guixiang Xie, Sebastian Mendoza Farias, Bas Teusink, and Johan van Heerden

Corresponding Author(s): Pranas Grigaitis, Vrije Universiteit Amsterdam

Review Timeline:

Submission Date:	May 5, 2022
Editorial Decision:	June 23, 2022
Revision Received:	July 19, 2022
Accepted:	July 21, 2022

Editor: Tal Korem

Reviewer(s): Disclosure of reviewer identity is with reference to reviewer comments included in decision letter(s). The following individuals involved in review of your submission have agreed to reveal their identity: Vahid Shahrezaei (Reviewer #3)

Transaction Report:

DOI: <https://doi.org/10.1128/msystems.00423-22>

June 23, 2022

Dr. Pranas Grigaitis
Vrije Universiteit Amsterdam
Amsterdam
Netherlands

Re: mSystems00423-22 (A computational toolbox to investigate the metabolic potential and resource allocation in fission yeast)

Dear Dr. Pranas Grigaitis:

Thank you for submitting your manuscript to mSystems. We have completed our review and I am pleased to inform you that, in principle, we expect to accept it for publication in mSystems. Please note, however, that a revised paper will be sent to reviewers, and acceptance will not be final until you have adequately addressed their comments.

Preparing Revision Guidelines

Sincerely,

Tal Korem

Editor, mSystems

Journals Department
Reviewer comments:

Reviewer #1 (Comments for the Author):

This is an excellent paper describing reconstruction of a proteome constrained genome-scale metabolic model for *S. pombe*. The paper is very well written and presents some novel and interesting results. I only have two questions that the authors should consider in a revision:

- 1) How were the *k_{cat}* determined? Were they taken from the Yeast8 model?
- 2) Interesting finding about the NGAM and that it is lower than for *S. cerevisiae*. Why is it like this? Are the authors sure about the P/O ratio. Also does *S. pombe* not have system 1 in the Ox-Phos and hereby has a higher ATP yield, and is there accounted for this. I find it strange that NGAM could be smaller - and if it really is then it will be very interesting to find out why as this could have a very large impact on biotechnology if this kind of mechanism could be transferred to *S. cerevisiae*.

Reviewer #2 (Comments for the Author):

The manuscript presents the development of a resource allocation model for the fission yeast *S. pombe*. First, a normal metabolic model is built through a genome-based reconstruction. This model is extended towards a proteome-constrained model according to the methodology previously published and applied to *S. cerevisiae* by the authors. An analysis of the proteome-constrained model indicates which physiological limitations shape *S. pombe* metabolism at different growth rates.

Overall it is an interesting study, and the results will be useful to the constraint-based metabolic modelling community. Both the basic GEM and the more extensive proteome-constrained model are relevant contributions to the available models and can be used in follow-up studies with various applications.

A major comment I have is about the uncertainty in model parametrization. Due to the use of for example *k_{cat}*-values, which are usually highly uncertain, this is more of an issue for pc-models than for GEMs. Also a confidence interval for the *k_{cat,ribo}* parameter should be determined. Generally, the authors don't investigate the uncertainty in their model analysis. I would like to see some indications at least how big is the uncertainty in the relevant model parameters, and how firm are the main results and conclusions despite such an uncertainty.

Despite my comment about the Sp GEM above, I think it can be useful if the authors discuss a bit more the relevance of the basic GEM they constructed. At the moment, this is mainly presented as a stepping stone to the pc-model, without clear indications where the authors see potential applications of the GEM. Obviously the GEM is in a way less predictive than the pc-model, otherwise the authors would not need to build the latter one, but in which situations could the GEM model still be useful? Also, are changes to the pc-model such as with the GAM discussed in lines 179-185, or the glucose-proton antiport taken back to the GEM somehow?

Smaller comments:

- Line 124 is only true for FBA predictions with the biomass reaction as an objective function (which is the most common of course but still should be mentioned)
- Switch to fermentation (line 127), does it really depend on the concentration of glucose, or rather on the growth rate? Provide a relevant experimental reference ideally.
- I'm not sure about the argument related to cytosolic proton availability in line 173-174. Yes, cytosolic protons are used to drive ATP synthase, but the key energy investment is in moving these protons from the mitochondrion to the cytosol, and thus what matters for ATP synthesis would actually be the proton balance in the mitochondrion, not in the cytosol, as the argument seems to imply.
- Since the unspecified protein fraction gets a major role, the paper should be a bit more explicit about what this constraint actually represents. At the moment it is not clear to me what is the physiological reasoning behind it.
- Line 231 - what were the flux predictions and this point and how were they different (and from which) experimental data?
- Line 239-240 - how was the speculation transformed into a "good flux prediction"?
- Line 419 - Do I understand correctly that 90% of maximal growth or lower was interpreted as "no growth"? This seems an overly strict high threshold, in other studies much smaller thresholds (e.g. $1e-5$ / h) are used. How does this influence the results in Figure 1 d-e?

Reviewer #3 (Comments for the Author):

The paper by Grigaitis et al propose a manually-curated high-quality genome-scale metabolic model in fission yeast. The authors also produce a proteome constraint based variant of the model, which is a first for *Pombe* I believe, following a recently developed model of this kind in budding yeast by the same group. Then they use these models to investigate metabolic potential

and resource allocation in this important model organism. This work highlights how metabolic knowledge can be transferred from better studied models such as budding yeast to less studied systems. Overall, this work is a great contribution to the community of both interested in constraint-based genome wide models of Eukaryotic metabolism and also to the large fission yeast community by providing them with some specific computational toolboxes. The paper is overall very well written. However, please find in the following some comments that if addressed could hopefully improve the paper even more. I note that these are mostly minor suggestions.

- Could you say more about the data shown in Fig. 1b, is this the most up-to-date measurement of cellular composition and is this defined for a specific growth condition?
- Not very clear what panel 1C is showing. How is this calculated? What is the dashed line and what is the solid line? Is the model doing well with predicated GAM values?
- The authors report 13.5% improvement with regards to true prediction rates compared to the 2012 model. Any insights what aspect of their model makes this improvement possible?
- What is the confidence interval of $\phi_0 = 0.08$ that was calculated from the data in Kleijn et al?
- In figure 4 the results of proteome allocation for maximum growth is compared with experimental data. Have the author's tried to produce results for the media used in Kleijn et al and compare the proteome allocation with the data in this paper?

Dear Dr. Korem,

On behalf of the Authors, I would like to thank you for the opportunity to revise our manuscript “A computational toolbox to investigate the metabolic potential and resource allocation in fission yeast”. Hereby we provide our responses (in blue) to the questions and suggestions, raised by the Reviewers.

Reviewer comments:

Reviewer #1 (Comments for the Author):

This is an excellent paper describing reconstruction of a proteome constrained genome-scale metabolic model for *S. pombe*. The paper is very well written and presents some novel and interesting results.

We appreciate the positive evaluation of our manuscript, and thank the Reviewer for their kind words.

I only have two questions that the authors should consider in a revision:

1) How were the k_{cat} determined? Were they taken from the Yeast8 model?

The k_{cat} values were manually collected from the BRENDA database for all the enzymes/enzyme complexes, annotated with Enzyme Commission (EC) numbers in the *pomGEM*. Indeed, most of these annotations come from the *Yeast8* model. The databases we used for collecting information required to construct the proteome-constrained model were reported in the *Methods* section, however, we now included an explicit statement on the collection of k_{cat} values (Lines 444-9) as follows:

“The detailed description of reconstruction of the proteome-constrained model of *S. pombe* is provided in the Supplementary Notes. We used the reference proteome of *S. pombe* from UniProt (The UniProt Consortium et al., 2021). The kinetic data (enzyme turnover values) were collected from the BRENDA database (Chang et al., 2021). For every enzymatic complex with an Enzyme Commission (EC) number, we queried the BRENDA database for a value from the wild-type enzymes. When available, values from *S. cerevisiae* or *S. pombe* were preferentially selected. Otherwise, the highest value for a wild-type enzyme in mesophilic (and close to growth conditions of *S. cerevisiae* or *S. pombe*) conditions was taken. When no k_{cat} value was available, we assumed $k_{cat} = 50 \text{ s}^{-1}$ as a default value (close to the median k_{cat} , Figure S2). If the experimentally determined k_{cat} value was lower than $2/3 \text{ s}^{-1}$, we set this value.”

2) Interesting finding about the NGAM and that it is lower than for *S. cerevisiae*. Why is it like this?

Here we would like to note that the NGAM value for *S. pombe* grown in glucose-limited conditions, reported in the literature, is not drastically different from the value reported for *S. cerevisiae* (current estimate for *S. c.* is $\sim 1 \text{ mmol gDW}^{-1} \text{ h}^{-1}$ vs. 0.66 – 0.83 for *S. p.*). It should be noted that the value for *S. c.* was recently re-evaluated using retenostats at extremely low dilution rates, while the data for *S. p.* from (de Queiroz et al. 1993) still come from chemostat measurements.

Are the authors sure about the P/O ratio. Also does *S. pombe* not have system 1 in the Ox-Phos and hereby has a higher ATP yield, and is there accounted for this.

In the manuscript, we have referred to the study by de Queiroz and colleagues, where the P/O ratio was measured experimentally. To our knowledge, this is the only study which has evaluated both the NGAM value, and the P/O ratio in *S. pombe*.

S. pombe, like *S. cerevisiae*, indeed lacks a proton-pumping complex I, , hence the similarly low P/O ratio , but little is known as to why the measured ratio in *S. pombe* is slightly higher than 1. However, we draw Reviewer's attention to the fact that the respiratory yield of biomass on glucose ($Y_{X/S}$) is in fact lower for *S. p.* compared to *S. c.* (0.43 vs. 0.52 $g (g \text{ glucose})^{-1}$).

I find it strange that NGAM could be smaller - and if it really is then it will be very interesting to find out why as this could have a very large impact on biotechnology if this kind of mechanism could be transferred to *S. cerevisiae*.

As we noted above, the difference to *S. c.* is not large, however, the Reviewer raises an excellent point. We speculate that a possible cause for lower maintenance requirements at [near-] zero growth might be linked to the ability of *S. p.* to produce metabolically-inactive spores under stress conditions. This is contrary to *S. c.* cells, which maintain reasonable metabolic activity even at extreme glucose limitation (see work from Daran-Lapujade lab at TU Delft).

Reviewer #2 (Comments for the Author):

The manuscript presents the development of a resource allocation model for the fission yeast *S. pombe*. First, a normal metabolic model is built through a genome-based reconstruction. This model is extended towards a proteome-constrained model according to the methodology previously published and applied to *S. cerevisiae* by the authors. An analysis of the proteome-constrained model indicates which physiological limitations shape *S. pombe* metabolism at different growth rates.

Overall it is an interesting study, and the results will be useful to the constraint-based metabolic modelling community. Both the basic GEM and the more extensive proteome-constrained model are relevant contributions to the available models and can be used in follow-up studies with various applications.

We thank the Reviewer for their evaluation, and we share the hope that these models will serve the community well.

A major comment I have is about the uncertainty in model parametrization. Due to the use of for example k_{cat} -values, which are usually highly uncertain, this is more of an issue for pc-models than for GEMs. Also a confidence interval for the $k_{cat,ribo}$ parameter should be determined. Generally, the authors don't investigate the uncertainty in their model analysis. I would like to see some indications at least how big is the uncertainty in the relevant model parameters, and how firm are the main results and conclusions despite such an uncertainty.

This is a very good comment and we acknowledge that we were too brief on explaining the kinetic parametrization of the *pcPombe* model, as the Reviewer #1 also indicated. Moreover, we agree that we have not tested how robust most of the parameters are. In order to cover different aspects of parametrization, we split the response into several parts.

- Kinetic parameters. We have appended our initial description on how we collected kinetic parameters, see response to the Reviewer #1. In many cases, no turnover values were

measured, and model contains several hundreds of metabolic complexes. Also, here we are limited to a single value for each enzyme that we either assume or take from the literature/databases. Therefore, a thorough sensitivity/perturbation analysis is not a viable option nor would give definitive results.

- Confidence intervals for the $k_{cat,ribo}$. As we also discuss in our response to a similar question by the Reviewer #3, we have not determined confidence intervals for both $k_{cat,ribo}$ and Φ_R^0 values – however, we now provide a 95% CI for the latter, see the response to Reviewer #3. We believe that estimating CI is not necessary for the $k_{cat,ribo}$: the analogous values for other major model organisms, say, *E. coli* or *S. cerevisiae* are always provided in low precision (20 and 10.5 $aa\ s^{-1}$, respectively). Moreover, the aforementioned measurements all come from rather old studies and have not been re-evaluated since. Thus, for both sake of consistency and reflecting the uncertainty, we do not present our estimate as a high-precision one (with an estimated CI).
- Robustness of prediction. We believe that the principal conclusions of the paper are rather robust, yet we agree with the Reviewer that testing perturbations to the model constraints/parameters would give more (and very valuable) insight into that. Unlike our recent study in *S. cerevisiae*, where we could test such scenarios against comprehensive datasets, a general paucity of metabolic data for *S. pombe* means that outcomes cannot readily be validated. We hope, of course, that ongoing experimental efforts aimed at quantifying *S. pombe* metabolism will yield such datasets in the coming years. .

Following the point raised on the ribosomal parameters: as discussed above, the model contains a lot of parameters with a high degree of uncertain (especially k_{cat} values). Yet, some of the major parameters (right-hand-sides of the constraint expressions, ribosomal parameters, and biomass composition) can act as buffers for the uncertainty by cancelling out the discrepancies caused by each other. As an example, we consider the maximal growth rate μ_{max} : for instance, an increase of the intercept Φ_R^0 value will result in a decreased maximal growth rate μ_{max} at glucose excess conditions. This effect would be cancelled with a more precise estimate of the minimal unspecified protein (UP) fraction, to the lower side (e.g. 0.25 \rightarrow 0.245). Therefore, we believe that this “self-balancing” and interdependency of the model outcome on several model parameters helps to confirm that the main observations are robust.

Despite my comment about the Sp GEM above, I think it can be useful if the authors discuss a bit more the relevance of the basic GEM they constructed. At the moment, this is mainly presented as a stepping stone to the pc-model, without clear indications where the authors see potential applications of the GEM. Obviously the GEM is in a way less predictive than the pc-model, otherwise the authors would not need to build the latter one, but in which situations could the GEM model still be useful?

Thank you for the suggestion. We wanted to emphasize that the GEM model is a useful resource for exploring the metabolic potential of *S. pombe*: we did so in the title, and mentioned some successful applications of GEMs in the Introduction:

“GEMs have successfully been applied in diverse settings, including the metabolic engineering of microorganisms (McAnulty et al., 2012; Mishra et al., 2018), studies of human diseases or disease causing pathogens (Beste et al., 2007; Branco dos Santos et al., 2017), drug development (Kim et al., 2011), and the investigation of interactions within microbial communities (Dukovski et al., 2021).”

Then we – although implicitly – recapitulated on that at the end of the Discussion:

“The two models we present here, therefore, are powerful tools that can be used to efficiently explore, in silico, S. pombe’s metabolic potential, to identify metabolic engineering targets, to design and optimize medium for different applications, and to study metabolic and physiological determinants of growth behaviour under different growth conditions.”

In line with the Reviewer’s suggestion, we have explicitly highlighted in the Discussion that GEM can be readily used for most of these applications:

*“~~The two models we~~ **pomGEM model** we present here, therefore, ~~is a~~ **are** powerful tools that can be used to efficiently explore, in silico, S. pombe’s metabolic potential, to identify metabolic engineering targets, to design and optimize medium for different applications. **These analyses can be complemented by,** ~~and to~~ **studies with pcPombe, directed at the** ~~γ~~ metabolic and physiological determinants of growth behaviour under different growth conditions.”*

Also, are changes to the pc-model such as with the GAM discussed in lines 179-185, or the glucose-proton antiport taken back to the GEM somehow?

The mode of glucose transport in the *pomGEM* is the same as in the *pcPombe*, proton symport. We clarified this in the text as follows (Lines 169-70):

*“<...>with a stoichiometry of 1:0.4 for glucose and protons, respectively (Hofer and Nassar, 1987). **We have thus corrected the stoichiometry of glucose import reactions in both pomGEM and pcPombe to reflect this.**”*

Smaller comments:

- Line 124 is only true for FBA predictions with the biomass reaction as an objective function (which is the most common of course but still should be mentioned)

Thank you for spotting this. We have explicitly mentioned the optimization objective in the revised version (Lines 124-5):

*“As a rule, FBA predictions will identify the metabolic strategy that leads to the highest biomass yield on the limiting nutrient **when optimization objective is maximization of growth rate.**”*

- Switch to fermentation (line 127), does it really depend on the concentration of glucose, or rather on the growth rate? Provide a relevant experimental reference ideally.

Thanks for pointing this out. What we actually meant was the glycolytic flux that is essential for the switch to fermentation, and glucose, of course, as its starting substrate. We have clarified this in the Main Text (Line 128):

*“In reality, cells will switch to fermentation, a lower ATP-yield strategy, ~~beyond a critical~~**when the glycolytic flux increases beyond a certain threshold.**”*

- I'm not sure about the argument related to cytosolic proton availability in line 173-174. Yes, cytosolic protons are used to drive ATP synthase, but the key energy investment is in moving these protons from the mitochondrion to the cytosol, and thus what matters for ATP synthesis would actually be the proton balance in the mitochondrion, not in the cytosol, as the argument seems to imply.

The Reviewer is correct about the *in vivo* situation of proton availability in different compartments. Yet we want to emphasize that the central assumption of flux balance analysis models – and one

introducing downstream issues we have to deal with – is that the fluxes *consuming* and *producing* every moiety must balance to zero. This way, as we argue in the Main Text, but also in the Supplementary Text, the availability of both mitochondrial and cytosolic protons are essential to determine the ATP synthesis capacity.

- Since the unspecified protein fraction gets a major role, the paper should be a bit more explicit about what this constraint actually represents. At the moment it is not clear to me what is the physiological reasoning behind it.

We apologize for being too implicit about what the UP minimum actually represents, and have expanded on this in the revised version (Lines 224-7):

“At very fast growth rates, instead of mitochondrial proteome capacity, the unspecified protein (UP) fraction (~~a proxy for the cytosolic proteome capacity~~), starts to limit growth. The UP is a collective term that aggregates all proteins that do not contribute directly to biomass synthesis (i.e., metabolically inactive proteins), into a single artificial protein of average composition and length. The minimal UP mass fraction is, therefore, a proxy for total cytosolic proteome capacity, which becomes an active constraint when the UP fraction in the proteome reaches the minimal value we estimated on the basis of proteomics data (Kleijn et al., 2022); at this minimal value, the cytosol is maximally filled with metabolically active proteins.”

- Line 231 - what were the flux predictions and this point and how where they different (and from which) experimental data?

Thank you for raising this point. We referred to the experimental data from de Jong-Gubbels (the chemostat experiment in Fig. 3), and we have specified that in the revised version. We have also briefly described the profile of these simulations in the revised Text (Lines 238-42):

*“We tested our claims by increasing the minimal UP fraction to the level that **sets the UP minimum to be hit at $\mu_{UP\ hit}^* = 0.16\ h^{-1}$ ($= \mu_{crit}$)**, and the glucose transporters are fully saturated and mitochondrial capacity constraint was relaxed. The flux predictions we acquired were considerably different from the experimental data of (de Jong-Gubbels et al., 1996): **a rapid increase in ethanol production was observed as the UP minimum was hit; and the maximal growth rate was $\mu_{max} = 0.18\ h^{-1}$. We concluded that the flux profile at the maximal growth rate $\mu_{max} = 0.18\ h^{-1}$ (which resembled experimental measurements at $\mu = 0.29\ h^{-1}$), was highly unlikely to be correct, and therefore we discarded such scenario.”***

- Line 239-240 - how was the speculation transformed into a "good flux prediction"?

We apologize for the confusing phrasing; what we meant was that our initial guess about the sequence of active proteome constraints seemed to be supported by proteomics data. We have rephrased this in the revised Text (Lines 249-50):

*“We constructed the *pcPombe* model with these observations from *S. cerevisiae* in mind, and since we achieved a good flux prediction, we argue that it is the active constraint under this growth regime.”*

- Line 419 - Do I understand correctly that 90% of maximal growth or lower was interpreted as "no growth"? This seems an overly strict high threshold, in other studies much smaller thresholds (e.g. $1e-5$ / h) are used. How does this influence the results in Figure 1 d-e?

The Reviewer's interpretation is correct, 90% of the WT growth rate was selected as the threshold. We acknowledge that the threshold is rather high, but would like to note that in most cases, the single-reaction knockouts were either completely silent (predicted growth rate approaching that of WT) or completely lethal (predicted growth rate is zero).

As suggested by the Reviewer, we also computed the true prediction rate when "essential" was assigned only in the case of complete lethality. A similar number of false predictions were both resolved and newly introduced by this, and thus the new true prediction rate was 74.7 -> 74.8%. We have highlighted this in the Main Text as follows:

"We benchmarked the pomGEM model by first predicting growth on single carbon sources (with only one false-negative, Figure 1d), lethal single-gene KOs (Figure 1e), and single-reaction KOs (Supplementary Table 3). For the latter, the fraction of true predictions was approximately 75%, a good improvement on the previously reported model (61.2%) (Sohn et al., 2012). In our study, we applied a rather stringent threshold for the viability of single-reaction KOs, considering the reaction essential if the predicted growth rate is below 90% of the wild-type value. We thus tested a different threshold (essential when the growth rate is zero) and arrived at effectively the same true prediction rate (74.7% vs 74.8% at zero growth threshold). This suggests that the overall performance of the pomGEM model in this regard is robust."

The interpretation of the Figure 1e (single *gene* knock-outs) is not affected by this, as the gene was assumed essential only when caused the predicted growth rate to be zero.

Reviewer #3 (Comments for the Author):

The paper by Grigaitis et al propose a manually-curated high-quality genome-scale metabolic model in fission yeast. The authors also produce a proteome constraint based variant of the model, which is a first for Pombe I believe, following a recently developed model of this kind in budding yeast by the same group. Then they use these models to investigate metabolic potential and resource allocation in this important model organism. This work highlights how metabolic knowledge can be transferred from better studied models such as budding yeast to less studied systems. Overall, this work is a great contribution to the community of both interested in constraint-based genome wide models of Eukaryotic metabolism and also to the large fission yeast community by providing them with some specific computational toolboxes. The paper is overall very well written.

We thank the Reviewer for the nice summary of our work and the enthusiasm towards the value of these models to the community.

However, please find in the following some comments that if addressed could hopefully improve the paper even more. I note that these are mostly minor suggestions.

- Could you say more about the data shown in Fig. 1b, is this the most up-to-date measurement of cellular composition and is this defined for a specific growth condition?

In the Manuscript, we have used the biomass composition, reported in the *SpoMBEL1693* model. It was aggregated from different literature sources, and, to our knowledge, is the only comprehensive overview of the biomass composition of *S. pombe* to the date. However, there is also additional information available for some of the biomass components. We quote from the Main Text:

"<...> *S. pombe* exhibits a constant protein density of $0.43 \text{ g (gDW)}^{-1}$ (de Jong-Gubbels et al., 1996) <...>"

We do not expect to observe major changes in macromolecular composition of the cells as a function of growth rate and/or conditions. However, a thorough assessment of biomass composition across conditions would be very beneficial for the future work on metabolism of *S. pombe*.

- Not very clear what panel 1C is showing. How is this calculated? What is the dashed line and what is the solid line? Is the model doing well with predicated GAM values?

We apologize for including an incomplete legend for the Panel 1C. Now we have specified what each of the lines means, Legend of Figure 1:

"c. Estimation of the GAM value. Glucose uptake flux was fixed to $1.0 \text{ mmol gDW}^{-1} \text{ h}^{-1}$ and the maximal specific growth rate μ (solid blue line) was predicted with varying GAM value. Growth yield on glucose $Y_{X/S}$ was computed based on the predicted specific growth rate. The target yield on glucose ($Y_{X/S} = 0.432 \text{ g biomass (g glucose)}^{-1}$, dashed horizontal line) was computed as an average of experimentally determined $Y_{X/S}$ from glucose-limited cultures with $D > 0.1 \text{ h}^{-1}$ (de Jong-Gubbels et al., 1996; de Queiroz et al., 1993; Uribe Larrea et al., 1997, 1993)."

- The authors report 13.5% improvement with regards to true prediction rates compared to the 2012 model. Any insights what aspect of their model makes this improvement possible?

We believe that the improvement is a combined outcome of several advancements throughout the years, combined to the model we compare the performance to. For instance, we used the latest reconstruction of *S. cerevisiae* metabolism, *Yeast8*, to reconstruct the *pomGEM*. Also, we used only one template model: usually, several template models are used to increase the coverage of the newly-created model, with a risk of introducing duplicate reactions. These are the most likely reasons behind a better true prediction rate.

- What is the confidence interval of $\phi_0 = 0.08$ that was calculated from the data in Kleijn et al?

We would like to correct that the estimated Φ_R^0 was around 0.05, and we have attempted the CI for this intercept. We refrained to report the CI because we were not sure about the measurements from the lowest- μ cultures from Kleijn *et al.*, which are considerably below the linear trend (see Fig 2C of the Main Text).

By excluding the data from cultures grown on glucose + tryptophan as carbon and nitrogen sources, respectively, we did estimate the 95% CI for the Φ_R^0 to be 0.0408-0.0522 and reported in the Main Text:

"First, we computed the fraction of "inactive" ribosomes $\Phi_R^0 \approx 0.05 \text{ (g g protein}^{-1}\text{)}$ (95% confidence interval, excluding the cultures grown with tryptophan as nitrogen source: 0.041-0.052) from the linear regression of the experimental data points (Figure 2c, black dashed line)."

- In figure 4 the results of proteome allocation for maximum growth is compared with experimental data. Have the author's tried to produce results for the media used in Kleijn et al and compare the proteome allocation with the data in this paper?

We thank the Reviewer for an excellent question. We indeed have done such experiments and compared the predicted proteome composition vs. experimental data for the conditions reported in Kleijn *et al.* (Xie & Grigaitis).

We have considered growth under either of the two regimes: operating at (i) nutrient-limited regime (amino acid transporter capacity limits maximal growth rate), or (ii) proteome-limited regime (batch culture, proteome capacity limits maximal growth rate). We used the *pcPombe* model to predict growth parameters (fluxes) and proteome composition. However, we could not select the “more plausible” scenario due to two reasons:

First, we could not validate the flux predictions as there were no flux measurements done in the study of Kleijn *et al.* Moreover, for both scenarios, the predicted proteome composition was very similar, with only minor differences both qualitatively and quantitatively. A major unifying trait of the predictions under both conditions was high undersaturation (computed as the ratio *measured vs. minimal predicted proteome fraction* $\ll 1$) for many enzymes. Unfortunately, due to the ambiguity of the two equivalent growth limiting regimes, we could not generate any meaningful insights without additional (flux) data, and therefore abandoned this comparison.

We would like to again thank the Reviewers for the assessment of our work, and hope that the revised manuscript will be suitable for publication in *mSystems*.

With warm regards,
Pranas Grigaitis

July 21, 2022

Dr. Pranas Grigaitis
Vrije Universiteit Amsterdam
Amsterdam
Netherlands

Re: mSystems00423-22R1 (A computational toolbox to investigate the metabolic potential and resource allocation in fission yeast)

Dear Dr. Pranas Grigaitis:

Thanks for responding to the comments by the reviewers, who were very positive about this work.

In light of the response, your manuscript has been accepted, and I am forwarding it to the ASM Journals Department for publication. For your reference, ASM Journals' address is given below. Before it can be scheduled for publication, your manuscript will be checked by the mSystems production staff to make sure that all elements meet the technical requirements for publication. They will contact you if anything needs to be revised before copyediting and production can begin. Otherwise, you will be notified when your proofs are ready to be viewed.

Publication Fees:

If you would like to submit a potential Featured Image, please email a file and a short legend to mSystems@asmusa.org. Please note that we can only consider images that (i) the authors created or own and (ii) have not been previously published. By submitting, you agree that the image can be used under the same terms as the published article. File requirements: square dimensions (4" x 4"), 300 dpi resolution, RGB colorspace, TIF file format.

We recognize that the video files can become quite large, and so to avoid quality loss ASM suggests sending the video file via <https://www.wetransfer.com/>. When you have a final version of the video and the still ready to share, please send it to mSystems staff at mSystems@asmusa.org.

Sincerely,

Tal Korem
Editor, mSystems

Pieter Dorrestein
Senior Editor, mSystems

Journals Department
Supplementary Text: Accept

Table S1: Accept

Table S2: Accept

Fig. S1: Accept

Table S3: Accept

Fig. S2: Accept